# Current Understanding of the Genetics and Molecular Mechanisms Regulating Wood Formation in Plants

**DOI:** 10.3390/genes13071181

**Published:** 2022-06-30

**Authors:** Min-Ha Kim, Eun-Kyung Bae, Hyoshin Lee, Jae-Heung Ko

**Affiliations:** 1Department of Plant & Environmental New Resources, Kyung Hee University, Yongin 17104, Korea; minha123@khu.ac.kr; 2Department of Forest Bio-Resources, National Institute of Forest Science, Suwon 16631, Korea; baeek@korea.kr

**Keywords:** biomass, secondary growth, vascular cambium, wood formation, xylem differentiation

## Abstract

Unlike herbaceous plants, woody plants undergo volumetric growth (a.k.a. secondary growth) through wood formation, during which the secondary xylem (i.e., wood) differentiates from the vascular cambium. Wood is the most abundant biomass on Earth and, by absorbing atmospheric carbon dioxide, functions as one of the largest carbon sinks. As a sustainable and eco-friendly energy source, lignocellulosic biomass can help address environmental pollution and the global climate crisis. Studies of *Arabidopsis* and poplar as model plants using various emerging research tools show that the formation and proliferation of the vascular cambium and the differentiation of xylem cells require the modulation of multiple signals, including plant hormones, transcription factors, and signaling peptides. In this review, we summarize the latest knowledge on the molecular mechanism of wood formation, one of the most important biological processes on Earth.

## 1. Introduction

Land plants can be divided into two main groups: vascular and non-vascular plants. The vascular system is one of the key factors that enabled plants to successfully settle on land about 470 million years ago. The vascular system not only transports water, nutrients, and signals throughout the plant but also serves as mechanical support that maintains the plant’s vertical growth, increasing access to sunlight. The secondary xylem consists of vessel, fiber, and tracheid cells, with a thick secondary cell wall (SCW) composed mainly of cellulose, hemicellulose, and lignin [1,2,3]. The secondary xylem is derived from the vascular cambium, which is a cylindrical secondary meristem. Wood formation is achieved through a series of cascading processes that include: xylem mother cell specification by vascular cambium cell division, the differentiation of these cells into xylem cells followed by cell expansion, secondary cell wall deposition, pit formation, and programmed cell death (Figure 1). Each step is elaborately coordinated by factors such as hormones, signal peptides, and transcription factors (TFs). Recently, due to the global climate crisis, interest in sustainable energy development using eco-friendly and renewable biomass is increasing. Woody biomass produced from wood formation processes offers an economic and sustainable feedstock for bioenergy production.

In this review, we summarize the current understanding of the genetic and molecular mechanisms regulating the development of the vascular cambium and xylem cell differentiation, discussing the regulation of the developmental processes of xylem cells. Understanding the molecular mechanisms of wood formation will help elucidate the evolution and history of vascular development in plants.

## 2. Vascular Cambium Cell Development and Xylem Cell Differentiation: Initiating Wood Formation

The growth and development of woody plants can be largely divided into primary and secondary phases. Primary growth occurs in the shoot apical meristem (SAM) within the ground and root apical meristem (RAM) in the basement. In the apical region of this meristem tissue, the procambium differentiates into primary phloem and primary xylem that comprise the primary vascular bundle. As the procambium expands to the interfascicular region, it develops into a ring to form vascular cambium [4]. Secondary growth refers to the production of wood by the vascular cambium. Vascular cambium divides to produce daughter cells, with secondary xylem formed toward the center and secondary phloem on the outside of the plant. Xylem mother cells can differentiate into several types of daughter cells. However, it is unknown how the cambium develops and how cell fate is determined via the cambium to the secondary xylem or secondary phloem.

Recently, considerable progress has been made in understanding the molecular mechanisms of cambium formation and development, based on research conducted in model plants such as *Arabidopsis* and poplar. These studies have revealed that cambium development is regulated by hormones, TFs, and signal peptides [5,6,7,8] (Figure 2a). A series of studies on the functional characteristics of *Arabidopsis* and poplar mutant plants have revealed genes that play an important role in the development of vascular cambium.

### 2.1. Regulation by Plant Hormones

Hormones control various functions such as plant growth, development, and stress resistance [9]. Many studies have reported that vascular cambium activity is regulated by several hormones, including auxins, cytokinins, ethylene, and brassinosteroids.

Auxins are hormones that function in plant growth and development and affect cellular processes such as cell division, expansion, and differentiation [10,11]. Auxins are highly expressed in the cambium cell layer and play an important role in vascular cambium initiation and development [12,13,14,15,16]. PtoIAA9 and AUX/IAA in *Populus tomentosa* interact with AUXIN RESPONSE FACTOR 5 (PtoARF5) to regulate vascular cambium cell division and secondary xylem development. Auxin signaling maximum leads to direct activation of CLASS III HOMEODOMAIN-LEUCINE ZIPPER III (HD-ZIP III) TFs and changes in cell type-specific transcriptomes that define xylem cell identities [16,17]. The PtoIAA9-PtoARF5 module can bind to the promoter of the HD-ZIP III genes *PtoHB7* and *PtoHB8*, which are involved in secondary xylem formation [18]. When the poplar HD-ZIP III gene *PtrHB4* was upregulated, cambium development was induced by enhanced expression of *PtrPIN1* [19]. The auxin-responsive module PaC3H17-PaMYB199 is associated with cambium cell division in poplar stems [20]. Knockdown mutants of two *Populus MADS-box* genes (*VCM1* and *VCM2*), which are modulators of auxin homeostasis specifically expressed in the vascular cambium, enhanced vascular cambium proliferation activity and subsequent xylem differentiation [21].

Cytokinins (CKs) play important roles in vascular cambium development together with auxins. Poplar cytokinin receptor genes *HISTIDINE KINASE 3* (*PtHK3a*) and *PtHK3b* are expressed at high levels in dividing cambium cells [22]. Inhibition of CK signaling by expression of *Arabidopsis AtCKX2* under the promoter of the birch *CRE1* gene reduced the number of cambium cells, whereas an increase in CK biosynthesis by expression of *Arabidopsis ISOPENTENYL TRANSFERASE 7* (*IPT7*) increased cambium cell division in transgenic poplars [14,22]. Auxin-regulated LONESOME HIGHWAY (LHW) and TARGET OF MONOPTEROS5 (TMO5)/TMO5-LIKE1 (T5L1) directly up-regulated the CK biosynthesis genes *LONELY GUY3* (*LOG3*) and *LOG4*, which are involved in the activation of vascular cell division [23]. *AT2G28510/DOF2.1*, which is a CK-dependent downstream target gene of LHW-TMO5/T5L1, controls vascular cell proliferation [24].

Brassinosteroids (BRs), ethylene (ET), and gibberellins (GAs) also promote vascular cambium division and secondary growth in trees. Mutations of both *BRI-LIKE 1* (*BRL1*) and *BRL3*, which are *Arabidopsis* vascular-specific BR receptors, resulted in reduced xylem formation and increased phloem development [25]. BRI1-EMS SUPPRESSOR 1 (BES1) was shown to be involved in xylem differentiation downstream of TDIF-TDR-GSK3s signaling [26]. Similarly, inhibition of BR synthesis resulted in decreased secondary xylem differentiation and SCW biosynthesis, whereas increased BR levels increased secondary growth in poplar [27]. Exogenous BR treatment or genetic complementation of the BR biosynthesis *DWARF* gene in BR-biosynthetic-mutant tomato with retardation of xylem development resulted in a complete recovery of xylem cell formation [28]. In contrast, overexpression of *GLYCOGEN SYNTHASE KINASE 3* (*SlGSK3*) or CRISPR/Cas9 knockout of *BRASSINOSTEROID-INSENSITIVE 1* (*SlBRI1*) to block BR signaling resulted in severely defective xylem differentiation and secondary growth in tomato [28]. The tonoplast membrane-localized auxin efflux carrier WALLS ARE THIN1 (SlWAT1) is directly activated by SlBRL1/2 in xylem precursor cells. Transposable element (TE)-mediated loss-of-function allele *Slwat1-copi* resulted in defects in secondary xylem development, with a reduced vessel element number in tomato. Secondary xylem formation of *Slwat1-copi* was completely recovered by genetic complementation of WAT1 function [29]. Cambium division was increased in ET-overproducing poplar, whereas it was decreased in ET-insensitive poplar [30]. *acs7-d*, an ET overproducing *Arabidopsis* mutant, showed enhanced cambial activity and reduced fiber cell wall development [31]. Transgenic poplar overexpressing the *GIBBERELLIN 20 OXIDASE 1* (*GA20ox1*) gene, which is involved in GA biosynthesis, was characterized by GA overproduction and cambium proliferation [32,33,34]. GA also induced vascular cambium differentiation and lignification when expressed downstream of WOX14 in *Arabidopsis* stems [35,36].

### 2.2. Regulation by Transcription Factors and Signal Peptides

Regulation via TFs and signal peptides is essential for vascular cambium development and xylem differentiation. CLAVATA3/EMBRYO SURROUNDING REGION (CLE) family peptides are 12–13 amino acids with two hydroxylated proline residues after processing and post-translational modifications [37]. CLE41/44 (a.k.a. TRACHEARY ELEMENT DIFFERENTIATION INHIBITORY FACTOR [TDIF]) is secreted by developing phloem cells and binds to the cambium cell receptor PHLOEM INTERCALATED WITH XYLEM (PXY) [38]. The TDIF-PXY module plays an important role in maintaining the cambium cell population via cell division and inhibiting xylem cell differentiation [39,40]. WUSCHEL HOMEOBOX RELATED 4 (WOX4), a downstream transcriptional regulator of the TDIF-PXY module, is specifically expressed in the cambium region, and is involved in the regulation of cambium activity [38,39,40,41,42]. Ectopic expression of *PttCLE41b* and other related *PttCLE41-like* genes causes a dwarf phenotype with loss of cell division orientation and defects in the patterning of vascular tissues [42]. *PtrCLE20* is expressed in xylem tissues and suppresses cambium activity in poplar [43]. *Populus WOX4*, *PttWOX4*, is specifically expressed in the cambial region during the growing season only. *PttWOX4a/b RNAi* transgenic poplar showed reduced vascular cambium width and secondary growth [42]. Auxin-dependent WOX4 regulation is achieved in xylem precursor cells with auxin maxima, which promote vascular cambium cell division in a non-cell-autonomous manner [16]. GLYCOGEN SYNTHASE KINASE3/SHAGGY-LIKE KINASE proteins (GSK3s), including ARABIDOPSIS BRASSINOSTEROID-INSENSITIVE2 (BIN2), BIN2-LIKE1 (BIL1), and BIL2 are downstream of the TDIF-PXY module and inhibit xylem differentiation via the suppression of BRI1–EMS–SUPPRESSOR1 (BES1) and BRASSINAZOLE RESISTANT 1 (BZR1) [26,44]. BIL1 is a key mediator linking the TDIF-PXY module with auxin-CK signaling for the maintenance of cambium activity [45]. BIL1 phosphorylates and activates MONOPTEROS/AUXIN RESPONSE FACTOR 5 (MP/ARF5), and activated MP promotes the expression of the CK signaling negative regulators ARABIDOPSIS RESPONSE REGULATOR 7 (ARR7) and ARR15, which suppress cambium cell division.

HD-ZIP III and NAC (NAM, ATAF and CUC) TFs have been reported to play important roles in wood formation. *PopREVOLUTA*, the closest poplar homolog gene to *Arabidopsis REVOLUTA* (*REV*), has been reported to be involved in vascular cambium initiation [46]. Poplar PtrHB5 and PtrHB7, which are the closest homologs of *Arabidopsis* CORONA and AtHB8, induce cambium activity and xylem differentiation during secondary growth [47,48]. *PtrHB7* is preferentially expressed in cambium tissues and is a direct target of the PtrIAA9-PtrARF5 module, inducing cambium activity and xylem differentiation [18,47,48]. *PtrHB4* is specifically expressed in shoot tips and in the early developmental stages of vascular tissue; PtrHB4-SRDX (PtrHB4 repressor) transgenic poplar showed defects in the secondary vascular system due to failure of interfascicular cambium formation [19].

The NAC TF family have a highly conserved N-terminal NAC domain, which is associated with nuclear localization, DNA binding, and homodimer and/or heterodimer formation with other NAC proteins [49]. Among the NAC TFs in *Arabidopsis*, VASCULAR-RELATED NAC DOMAINs (VNDs) are master regulators of xylem differentiation [50,51]. NAC SECONDARY WALL THICKENING PROMOTING FACTOR 1 and 3 (NST1 and 3) are involved in stem fiber differentiation [52,53]. VND1, VND2, and VND3 contribute to xylem vessel formation during seedling development [54]. Poplar VNDs and NST1/3 are also involved in xylem differentiation [55,56,57]. VND6 and VND7, in particular, have been studied as master switches of xylem differentiation in metaxylem and protoxylem cells, respectively [50,58,59]. VND6 and VND7 upregulate xylem vessel cell differentiation-related genes such as SCW biosynthesis genes and programmed cell death (PCD) genes [58,60,61,62]. VND6 and VND7 directly regulate MYB46 and MYB83, which are master regulators of SCW biosynthesis [63,64]. XYLEM DIFFERENTIATION AND ALTERED VASCULAR PATTERNING (XVP) is a negative regulator of the TDIF-PXY module and fine-tunes TDIF signaling for xylem differentiation via interacting with the PXY-BAK1 (BRI1-ASSOCIATED KINASE 1) receptor complex [65].

However, some NAC TFs function as negative regulators of xylem differentiation. In *Arabidopsis*, a *XYLEM NAC DOMAIN 1* (*XND1*) mutant showed improved xylem differentiation [66], and poplar *XND1* or *PopNAC122* (XND1 ortholog) overexpressing transgenic *Arabidopsis* had a decreased wood formation phenotype [67]. Expression of C-terminal truncated *VND-INTERACTING2* (*VNI2*) under the control of the *VND7* promoter resulted in inhibition of xylem vessel development in *Arabidopsis* roots and aerial organs [68]. Recently, the authors of [69] demonstrated that VND7 downregulated *VNI2* expression and that MYB83, a downstream target of VND7, upregulated *VNI2* expression.

## 3. Cell Expansion: Determining the Final Shape and Size of the Xylem Cell

Wood formation is achieved through a series of processes, including cell expansion, SCW deposition, and programmed cell death after the determination of the fate of daughter cells formed by vascular cambium division. The cell expansion step determines the final shape and cell size of the xylem and takes place in the primary cell wall (PCW) formation stage before the SCW is formed.

The *EXPANSIN* family comprises four subfamilies: α-expansin (EXPA), β-expansin (EXPB), expansin-like A (EXLA), and expansin-like B (EXLB). EXPA and EXPB bind to xyloglucan and xylose, respectively, and break noncovalent bonds between other cell wall components, resulting in cell wall loosening [70,71,72]. XYLOGLUCAN ENDOTRANSGLUCOSYLASE/HYDROLASE (XTH) is involved in cell wall loosening and remodeling by catalyzing the hydrolysis and reconnection of xyloglucans [73,74]. PECTIN METHYLESTERASE (PMEs) and PECTIN ACETYLESTERASE (PAEs) regulate cell wall loosening through pectin modification [75,76,77]. *Populus ETHYLENE RESPONSE FACTOR 85* (*PtERF85*) is expressed in the phloem, cambium cells, and expanding xylem but not in mature xylem cells. The ectopic expression of *PtERF85* reduced wood density and SCW thickness of xylem fibers in association with decreased expressions of SCW biosynthesis genes, but the diameter of fiber cells increased [78]. Thus, PtERF85 activates xylem cell expansion, but prevents SCW formation, suggesting that PtERF85 contributes to the transition of fiber cells from elongation to secondary cell wall deposition.

After cell wall expansion, SCW deposition occurs. Several excellent reviews have been reported recently on this subject, so please see [79,80,81,82].

## 4. Pit Formation: Decorating SCW

Cellulose microfibrils are the main components of the plant cell wall and physically restrict cell expansion, causing anisotropic cell growth. Cellulose microfibrils are synthesized at the plasma membrane outer surface by CESA complexes. The orientation of cellulose microfibrils is directed by cortical microtubules [83,84,85]. Thus, the pattern of cortical microtubule alignment affects the overall deposition pattern of cellulose microfibrils, which determines the shape of plant cells. Microtubule-associated proteins are important for regulating the dynamics and interactions of cortical microtubules. Plant-specific conserved microtubule-associated proteins such as ROP-INTERACTIVE CRIB MOTIF-CONTAINING PROTEIN1 (RIC1) and SP1-LIKE2 (SPL2) are involved in regulating the movement of transverse cortical microtubules [86,87,88].

Unique deposition patterns of the secondary cell walls of xylem cells, such as spiral and reticular patterns, are also determined by cortical microtubule alignment. During xylem cell differentiation, transverse cortical microtubules are rearranged into bundled or pitted patterns to direct secondary cell wall patterns [89]. MICROTUBULE-ASSOCIATED PROTEINS 70-5 (MAP70-5), MAP65, AUXIN-INDUCED IN ROOT CULTURES 9 (AIR9), CELLULOSE SYNTHASE-INTERACTIVE PROTEIN1 (CSI1), and MICROTUBULE DEPLETION DOMAIN 1 (MIDD1), which are members of a plant-specific microtubule-associated protein family, are involved in regulating secondary cell wall patterns [90,91,92]. Plasma membrane domains are formed by local activation of Rho-like GTPase RHO-RELATED PROTEIN FROM PLANTS 11 (ROP11) by RHO GUANYL-NUCLEOTIDE EXCHANGE FACTOR 4 (ROPGEF4) and ROP GTPASE-ACTIVATING PROTEIN 3 (ROPGAP3) [93]. Activated ROP11 interacts directly with MIDD1 and anchors it to the plasma membrane domain [91]. Kinesin-13A interacts with MIDD1 through a C-terminal coiled-coil domain and functions in microtubule degradation through the active ROP-MIDD1 cascade [94,95,96]. Loss of *CORTICAL MICROTUBULE DISORDERING1* (*CORD1*) and *CORD2*, which encode microtubule-associated proteins, resulted in an irregular secondary cell wall with small pits in the xylem cells [97]. BOUNDARY OF ROP DOMAIN1 (BDR1) and WALLIN (WAL) localize to pit boundaries and mediate an ROP-actin pathway that shapes pit boundaries. WAL interacts with F-actin and promotes actin assembly at pit boundaries. BDR1 interacts with WAL as an ROP effector [98].

## 5. Programmed Cell Death: Finalizing Xylem Differentiation

After SCW biosynthesis, programmed cell death (PCD) occurs as the final step in the formation of mature xylem cells. PCD is usually initiated by central vacuole rupture during tracheary element (TE) differentiation, in which hydrolytic enzymes such as proteinases and nucleases are released [99,100].

*BIFUNCTIONAL NUCLEASE1* (*BFN1*) encodes a bifunctional nuclease I enzyme with both RNase and DNase activities [100]. *XYLEM CYSTEINE PROTEASE1* (*XCP1*) and *XCP2*, which have xylem-specific expression [101], are papain-like cysteine proteases located in the vacuole that control micro-autolysis in intact vacuoles or mega-autolysis of all cell materials in tonoplasts after rupture of the cell [102]. *Arabidopsis* METACASPASE (MC) family proteins induce cell death [103,104]. Among them, *Arabidopsis AtMC9* has a developing xylem-specific expression pattern [60,105], and the poplar homolog of *AtMC9* is upregulated during xylem maturation [106]. PCD-related genes such as *XCP1*, *XCP2*, and *AtMC9* are direct targets of VND6 and VND7 [60,107]. This indicates that VND6 and VND7 regulate PCD in TE, in addition to SCW, suggesting that PCD is an essential part of the TE maturation program.

PCD is an important step in wood formation, involving the breakdown of all cellular contents except the cell wall. Although our understanding of PCD related to wood formation is insufficient, we are optimistic that future studies will provide insights into wood formation. 

We summarized the recent wood formation studies described here in Table 1.

## 6. Conclusions and Future Perspectives

In recent years, advances in molecular biology technologies and bioinformatics have facilitated in-depth research into the regulatory networks governing vascular cambium development, cell differentiation, and secondary xylem formation. Wood formation is achieved through systematic and intimate processes controlled by a wide variety of factors, including hormones, small peptides, and transcriptional regulators. However, much is still unknown about the detailed mechanisms of wood formation, including cell fate determination and vascular cambium differentiation from procambium. The study of wood formation is difficult because the vascular cambium cells are embedded beneath the layers of other tissues, making them difficult to access. Additionally, in general, woody plants have a very long vegetative growth period, being large in size. The contribution of various cell types to wood formation is not well understood; until now, only tissue-level analysis was possible. Very recently, single-cell RNA-seq enabled the analysis of the wood formation control network at the single-cell level [108,109]. In the future, new information on the molecular mechanisms involved in wood formation will likely be obtained through studies using the latest technologies, which are rapidly being developed and applied.

## Figures and Tables

**Figure 1 genes-13-01181-f001:**
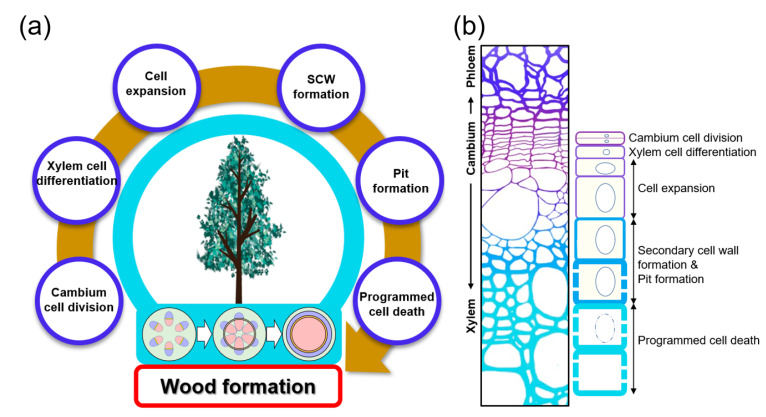
**Wood formation in plants.** (**a**) Simplified process of wood formation. Wood formation is initiated through cell divisions within the cylindrical vascular cambium layer, formed from the procambium. Then xylem cell differentiation, cell expansion, secondary cell wall (SCW) and pit formation, and programmed cell death (PCD) follow. Vascular cambium formation from procambium is shown in the stem cross-sections from below a tree: xylem (red), phloem (blue), and the cambium (yellow). (**b**) Stem cross-section and cell diagram for each stage of wood formation. The xylem is formed through the vascular cambium cell division and xylem cell differentiation, cell expansion, secondary cell wall formation and pit formation, and programmed cell death.

**Figure 2 genes-13-01181-f002:**
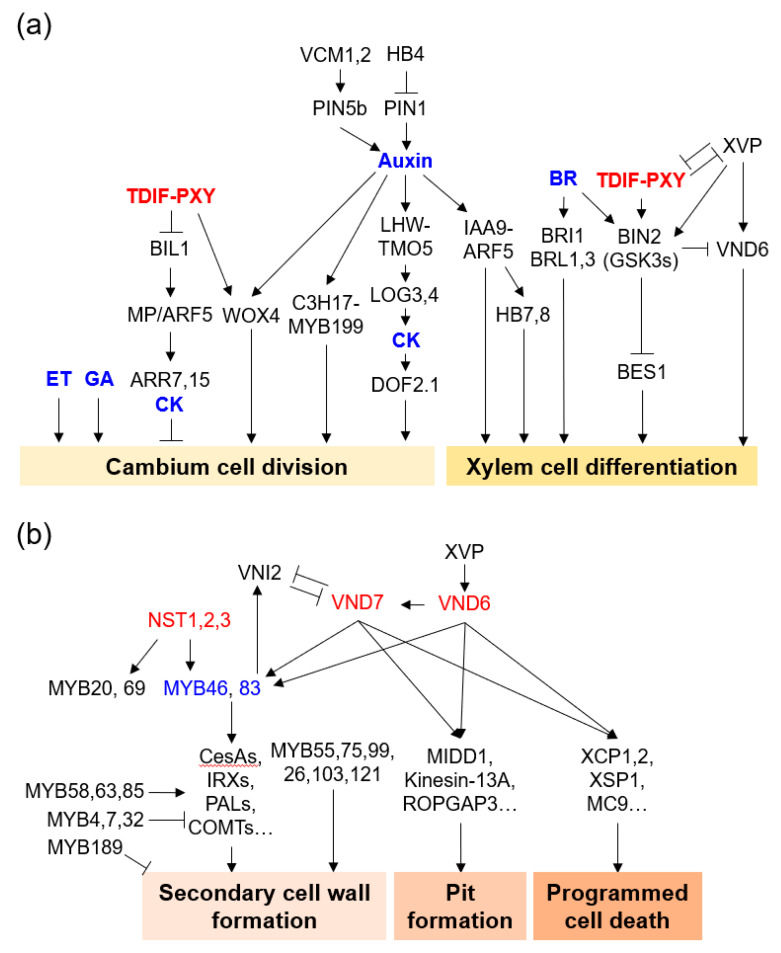
Molecular regulatory network of xylem cell formation. (**a**) Vascular cambium cell division and xylem cell differentiation. (**b**) Secondary cell wall formation, pit formation, and programmed cell death in xylem cell formation. Representative genes were depicted with hormones (blue). Genes with essential roles are described in the text.

**Table 1 genes-13-01181-t001:** Summary of recent wood formation studies.

Gene Name	Function	Studied Plant Species	Reference
** *Regulation by plant hormones* **			
*PtoIAA9*-*PtoARF5* module	Regulate vascular cambium cell division and secondary xylem development	*Populus tomentosa*	[16,17,18]
*PtoHB7* and *PtrHB8*	Secondary xylem cell differentiation, direct target of *PtoIAA9*-*PtoARF5* module	*Populus tomentosa*	[18]
*PtrHB4*	Increase cambium development	*Populus trichocarpa*	[19]
*PaC3H17*-*PaMYB199* module	Regulate cambium cell division	*Populus trichocarpa*	[20]
*VCM1* and *VCM2*	Knockdown mutant enhanced vascular cambium proliferation and xylem differentiation	*Populus deltoides × P. euramericana*	[21]
*AtCKX2*	Reduce cytokinin signaling and cambium cell growth	*P. tremula × tremuloides*	[22]
*IPT7*	Key enzymes in the biosynthesis of major bioactive cytokinins, increase cambium cell division	*Populus tremula × tremuloides*	[14]
*LHW*-*TMO5* module	Induce vascular cell proliferation	*Arabidopsis thaliana*	[23,24]
*LOG3* and *LOG4*	Activate vascular cell division	*Arabidopsis thaliana*	[23]
*DOF2.1*	Cytokinin-dependent vascular cell proliferation	*Arabidopsis thaliana*	[24]
*BRL1* and *BRL3*	Reduce xylem formation and increase phloem development	*Arabidopsis thaliana*	[25]
*BES1*	Promotes xylem differentiation from procambial cells	*Arabidopsis thaliana*	[26]
*SlGSK3*	Reduce xylem differentiation	*Solanum lycopersicum*	[28]
*SlBRI1*	Promote xylem differentiation	*Solanum lycopersicum*	[28]
*SlWAT1*	Increase secondary xylem development	*Solanum lycopersicum*	[29]
*ACS7*	*acs7-d* mutant enhanced cambium activity and reduced fiber cell wall development	*Arabidopsis thaliana*	[31]
*GA20ox1*	GA over production and cambium proliferation	*P. alba × P. tremula var. glandulosa*	[33,34]
** *Regulation by transcription factors and signal peptides* **			
*CLE41*/*44*(*TDIF*)-*PXY* module	Maintaining the cambium cell population via cell division and inhibiting xylem cell differentiation	*Arabidopsis thaliana,* *Populus tremula × P. tremuloides*	[39,40,41]
*WOX4*	Regulation of cambium activity	*Arabidopsis thaliana,* *Populus tremula L. × P. tremuloides*	[16,42]
*PttCLE41b* and *PttCLE41*-*like*	Reduce cell division and defect vascular tissue pattern	*Populus tremula L. × P. tremuloides*	[41,42]
*PtrCLE20*	Reduce cambium cell activity	*Populus trichocarpa*	[43]
*BIN2*, *BIL1* and *BIL2*	Inhibit xylem differentiation by suppression of BES1 and BZR1	*Arabidopsis thaliana*	[26,44,45]
*ARR7* and ***ARR15***	Supress cambium cell division	*Arabidopsis thaliana*	[45]
*PopREVOLUTA*	Involved in vascular cambium initiation	*Populus trichocarpa*	[46]
*PtrHB5* and *PtrHB7*	Induce cambium activity and xylem differentiation	*Populus tomentosa,* *Populus alba × P. tremula,* *Populus × euramericana*	[18,47,48]
*VND*s (*VND1–7*)	Involved in xylem differentiation	*Arabidopsis thaliana,* *Populus trichocarpa,* *Dactylis glomerata L*	[54,55,56,57,62]
*MYB46* and *MYB83*	Master regulators of SCW biosynthesis	*Arabidopsis thaliana*	[63,64]
*XVP*	Negative regulator of the TDIF-PXY module	*Arabidopsis thaliana*	[65]
*XND1* and *PopNAC122*	Mutant showed improved xylem differentiation	*Arabidopsis thaliana*	[66]
*VNI2*	Inhibition of xylem vessel development	*Arabidopsis thaliana*	[69]
** *Cell expansion* **			
*XTH*	Involved in cell wall loosening	*Populus tremula × tremuloides*	[74]
*PME*s and *PAE*s	Regulate cell wall loosening through pectin modification	*Populus trichocarpa*	[77]
*PtERF85*	Contributes to the transition of fiber cells from elongation to secondary cell wall deposition	*Populus tremula L. × P. tremuloides*	[78]
** *Pit formation* **			
*RIC1* and *SPL2*	Regulating the movement of transverse cortical microtubules	*Arabidopsis thaliana*	[88]
*MAP70-5*, *MAP65*, *AIR9*, *CSI1* and *MIDD1*	Involved in regulating secondary cell wall patterns	*Arabidopsis thaliana*	[92]
*ROP11*	Direct formation of cell wall pits in metaxylem vessel cells through interaction with cortical microtubules	*Arabidopsis thaliana*	[93]
*ROPGEF4*	Regulates the formation of ROP-activated domains	*Arabidopsis thaliana*	[93]
*ROPGAP3*	Positively regulates pit formation	*Arabidopsis thaliana*	[93]
*Kinesin-13A*	Microtubule degradation through the active ROP-MIDD1 cascade	*Arabidopsis thaliana*	[94,95,96]
*CORD1* and *CORD2*	Mutants resulted in an irregular secondary cell wall with small pits in xylem cells	*Arabidopsis thaliana*	[97]
*BDR1*	Interacts with F-actin and promotes actin assembly at pit boundaries.	*Arabidopsis thaliana*	[98]
*WAL*	Interacts with WAL as an ROP effector	*Arabidopsis thaliana*	[98]
** *Programmed cell death* **			
*BFN1*	Involved in nucleic acid degradation to facilitate nucleotide and phosphate recovery during senescence.	*Zinnia elegans*	[100]
*XCP1* and *XCP2*	Function in micro-autolysis within the intact central vacuole	*Arabidopsis thaliana*	[102]
*AtMC9*	Induce xylem cell death	*Arabidopsis thaliana*	[104]

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
