# Peer review of "Current Understanding of the Genetics and Molecular Mechanisms Regulating Wood Formation in Plants"

_genes, 2022, doi:10.3390/genes13071181_

Round 1

Reviewer 1 Report

In this study, the authors have reviewed the latest knowledge on the molecular mechanism of wood formation. The manuscript is well written, and I enjoyed reading the manuscript. However, minor concerns should be addressed before the manuscript can progress further. I hope the authors find these comments helpful in maximizing the impact of their work.

1- In the abstract, I would suggest limiting the background to two sentences and including the study's aims and implications of the review's findings.

2- In the introduction, I suggest including what motivated the authors to conduct this review study and why this topic is interesting. 

3- In the conclusions and future perspective, the authors should expand this section to highlight the review's significance by including the most compelling questions raised. 

Author Response

Dear reviewer 1,

Thanks for your excellent review comments.

Due to the format of response letter, please take a look at attached pdf file.

Thanks again for your efforts. With my best regards.

Reviewer 2 Report

Comments to the authors:

The processes have been described and mechanisms have been well discussed in the current review “Current understanding of the genetics and molecular mechanisms regulating wood formation in plants”. However, the detailed mechanism of wood formation including the molecular level mechanism can be added. In addition, I would recommend adding a relevant table citing most recent studies.

Author Response

Dear reviewer 2,

Thanks for your excellent review comments.

Due to the format of response letter, please take a look at attached pdf file.

Thanks again for your efforts. With my best regards.

Author Response

Dear reviewer 3,

Thanks for your excellent review comments.

Due to the format of response letter, please take a look at attached pdf file.

Thanks again for your efforts. With my best regards.

Round 2

Reviewer 3 Report

 Comment #9: line 199 as “Recently, Ailizati et al. [69] demonstrated that” , but  'Ailizati et al. ' can not been found in the  revised manuscript, please check!